# Assessment of the Impact on 20 Pelagic Fish Species by the Taiwanese Small-Scale Longline Fishery in the Western North Pacific Using Ecological Risk Assessment

**DOI:** 10.3390/ani12162124

**Published:** 2022-08-19

**Authors:** Kwang-Ming Liu, Lung-Hsin Huang, Kuan-Yu Su

**Affiliations:** 1Institute of Marine Affairs and Resource Management, National Taiwan Ocean University, Keelung 20224, Taiwan; 2George Chen Shark Research Center, National Taiwan Ocean University, Keelung 20224, Taiwan; 3Center of Excellence for the Oceans, National Taiwan Ocean University, Keelung 20224, Taiwan

**Keywords:** tuna, billfish, sharks, productivity, susceptibility, fishing impact

## Abstract

**Simple Summary:**

In this study, a semi-quantitative ecological risk assessment (ERA) was used to evaluate the ecological risks of fishing impact on 20 pelagic fish species by the small-scale longline fisheries in the western North Pacific Ocean. More than 2.38 million individual landing records at Nangfangao and Hsinkang fishing ports, eastern Taiwan from 2001–2021 were used in this study. The productivity was estimated based on the mean ranking (high, median, and low) of seven life history parameters and the susceptibility was calculated by the multiplication of the catchability, selectivity and post-capture mortality. The ERA results indicated sharks have higher ecological risk than those of tunas and billfishes, except yellowfin tuna (*Thunnus albacares*). The shortfin mako shark (*Isurus oxyrinchus*) and dusky shark (*Carcharhinus obscurus*) have the highest risk. Yellowfin tuna, other shark species, and sailfish (*Istiophorus platypterus*) have medium risk. While the striped marlin (*Kajikia audax*), and albacore tuna (*T. alalunga*) have the lowest risk. Although ERA cannot replace the conventional stock assessment methods that can produce solid management information on catch and effort, yet it can provide useful information for precautionary management measures.

**Abstract:**

Ecological risk assessment (ERA) has been applied on assessing the relative risk of bycatch species in recent years. ERA index is calculated by productivity of species and susceptibility of fisheries on fish species. In this study, a semi-quantitative method was used to evaluate the risks of exploitation for 20 pelagic fish species by the small-scale longline fisheries in the western North Pacific Ocean. The productivity was estimated based on the ranking (high, median, and low) of seven life history parameters. The susceptibility was calculated by the multiplication of the catchability, selectivity and post-capture mortality. The ERA results indicated the risks of sharks are higher than those of tunas and billfishes, except yellowfin tuna (*Thunnus albacares*). The shortfin mako shark (*Isurus oxyrinchus*) and dusky shark (*Carcharhinus obscurus*) have the highest risk. Other shark species, yellowfin tuna, and sailfish (*Istiophorus platypterus*) have medium risk. While the striped marlin (*Kajikia audax*), and albacore tuna (*T. alalunga*) have the lowest risk. Stock assessment and rigorous management measures such as catch quota and size limit are recommended for the species in high or medium ecological risk and a consistent monitoring management scheme is suggested for those in low ecological risk.

## 1. Introduction

The Taiwanese small-scale tuna longline fishing vessels (<100 gross tonnages, GRT) operating in the coastal and offshore waters of Taiwan, western North Pacific are mainly based on Nanfngao fishing port in the northeastern Taiwan and Hsinkang fishing port in the southeastern Taiwan. According to the sales records, annual landings of Nanfangao fishing port ranged from 6656 tons in 2006 to 2863 tons in 2014; those of Hsingkang fishing port ranged from 2723 tons in 2007 to 1152 tons in 2018 during the period of 2001–2021. The fishing vessels of these two fishing ports switch target species by season with altering gear configuration, fishing area, and baits. Most vessels target the dolphin fish, *Coryphaena hippurus* from March to June, the Pacific bluefin tuna, *Thunnus orientalis*, from May to June, other tunas *Thunnus* spp. from March to July, billfishes from March to October, and some vessels target pelagic sharks from October to next March [1]. Other tunas include the yellowfin tuna, *T. albacares*, the albacore tuna, *T. alalunga*, and the bigeye tuna, *T. obesus.* Billfishes include the swordfish, *Xiphias gladius*, the blue marlin, *Makaira nigricans*, the black marlin, *Istiompax indica*, and the sailfish, *Istiophorus platypterus*. Pelagic sharks mainly include 11 species, namely, the bigeye thresher *Alopias superciliosus*, pelagic thresher, *A. pelagicus*, silky shark, *Carcharhinus facilformis*, spinner shark, *C. brevippina*, dusky shark, *C. obscurus*, oceanic whitetip, *C. longimanus*, sandbar shark, *C. plumbeus*, scalloped hammerhead, *Sphyrna lewini*, smooth hammerhead, *S. zygaena*, blue shark, *Prionace glauca*, and shortfin mako shark, *Isurus oxyrinchus*. Of these species, the dolphin fish and yellowfin tuna are the two major species in terms of catch in number and weight. In addition, other teleost species such as Scombridae including mackerel *Scomber* spp. and Spanish mackerel *Scomberomorus* spp. were also caught but these species were not examined in this study because their catch in number and individual weight data were not available.

Conventional single-species models have commonly been used in fish stock assessment [2] and have been applied to sharks [3,4,5,6,7,8,9,10,11,12], tuna [13,14,15], and billfishes [16,17,18] in the western North Pacific. In addition to single-species approach, multi-species approach based on life history parameters has been developed for 57 teleost fish [19], 39 shark [20] and 38 skate and ray species [21]. However, none of these studies have assessed the relative risk of pelagic fish species in this region which can be used for ecosystem-based management.

In recent years, the ecological risk assessment (ERA) method such as productivity-susceptibility analysis (PSA) has been proposed for ecosystem-based fisheries management, a group of marine animals can be analyzed based on existing biological and fishery information [22]. The method is calculated by combining the productivity of animals and the susceptibility to fisheries to assess the relative vulnerability of stocks [23]. The relative risk, priority of management and conservation of animals can be identified based on the ERA results. The ERA method has been used to evaluate the vulnerability of the Atlantic tuna fisheries by the United States and European Union [24], Alaska demersal fishery [23], and Australian trawl fishery [25] as well as the risk of ecological animals of tuna fisheries by various tRFMOs [26,27,28,29,30].

Although the risk of species caught in the western and central Pacific Ocean (WCPO) tuna fisheries has been assessed based on ERA, such an approach has never been applied to multi-species management in the western North Pacific Ocean, apart from in recent studies by Lin et al. [31] and Liu et al. [1]. Lin et al. [31] used a semi-quantitative PSA on identifying fishery interactions with 52 species caught by six fishing gears in the eastern Taiwan waters, while Liu et al. [1] assessed the vulnerability of 11 pelagic sharks by the Taiwanese longline fishery using an integrated ERA, which coupled ERA with the IUCN Red List category, the body weight variation trend, and the inflection point of population growth curve. However, the specific impact on pelagic fish species including tuna, billfishes, and sharks by the Taiwanese small-scale longline fishery has not been examined. Hence, the objective of this study is to assess the impact of the Taiwanese small-scale tuna longline fishery on the 20 pelagic fish species in the western North Pacific Ocean by using a semi-quantitative ecological risk assessment method. It is hoped that the results derived from this study can be used as a reference for implementing ecosystem-based management and prioritizing the conservation measures of pelagic species in this region.

## 2. Materials and Methods

### 2.1. Study Area and Species

This study covers the western North Pacific ranging from 20° N to 30° N, and from 120° E to 140° E, the conventional fishing ground for the small-scale longline vessels based in eastern Taiwan (Figure 1). Twenty pelagic fish species including tunas, billfishes, and sharks were analyzed in this study (Table 1).

### 2.2. Source of Data

The sales data of two major fishing ports of Taiwanese small-scale longline fishery–the Nanfanao and Hsinkang fishing ports—were used in the analysis as the landings of these two fishing ports comprised of 80% of the catch of Taiwanese longline fishery in the study area of the western North Pacific. Each of the individual tuna, billfishes, and sharks landed at the Nanfangao and Hsinkang fishing ports, eastern Taiwan, was weighed before auction. Therefore, we were able to collect the species-specific whole weight data of each individual from the sales records. These data, including 20 pelagic fish species from 2001–2021, were used to estimate the species-specific catch in number, the catch composition, and the mean and median weight. The length was estimated from the individual weight based on existing length-weight relationships. The age of each individual was then estimated by substituting the length into the growth equation of each species from the literature [Appendix A]. As most blue sharks that landed at the Nanfangao fishing port were not sold in a regular fish market channel, only weight without catch in number data were available. A mean whole weight of 32 kg reported by the on-board scientific observers of Taiwanese longline vessels in the North Pacific was used to estimate the catch in number for blue shark. Various length measurements have been applied to different fish species. For example, total length (TL) was commonly used in pelagic sharks, fork length (FL) in tunas, and eye-fork length (EFL) or lower jaw fork length (LJFL) in billfishes. For consistency, EFL was used for all billfish species in this study.

In addition to the 20 species examined in this study, the dolphinfish, *Coryphaena hippurus*, is the most abundant pelagic fish species caught by the Taiwanese small-scale longline fishery in the western North Pacific. Its catch in number was much more than other pelagic fish species and the information was not available, and the dolphinfish was usually caught by shallow set longline fishery, which was different from those targeting tuna, billfishes, and sharks. Therefore, the dolphinfish was excluded in this study.

### 2.3. ERA

The ERA method considers the productivity of fish and their susceptibility to fisheries. The estimation of the productivity score of 20 pelagic fish species followed Hobday et al. [32] based on seven life history parameters. The rank score (R = 1–3) was assigned to each of the life history parameters and the mean value of these scores represented the productivity index (P) for each species. The higher productivity score meant higher productivity (less risk) of a fish species. The seven life history parameters and their productivity scores are categorized based on the criteria in Table 2.

Susceptibility (S) is the impact of a fish species from fisheries and is estimated by the multiplication of the probabilities of the following three parameters: (1) catchability, the species composition (percentage of catch in weight) of 20 species based on the landing data from 2001–2021 at Nanfangao and Hsinkang fishing ports, eastern Taiwan; (2) selectivity, the ratio of age range of catch and longevity, can be expressed as: (A_max_c_–A_min_c_)/A_max_, where A_max_c_ is the maximum age at catch, A_min_c_ is the minimum age at catch, A_max_ is the longevity [33]; (3) post-capture mortality, including the mortality of retention and after discard or live release. The post-mortality was referred to Cortés et al. [30] that estimated from the observed data of US far sea fishery. Susceptibility was estimated from the multiplication of the aforementioned three parameters. When the productivity and susceptibility were estimated, as these two were in different scale, a modified standardized Euclidean distance (D) from [29] was calculated:(1)$$D=\sqrt{{\left(\frac{P-2.5}{1.5}\right)}^{2}+{\left(\frac{S-0}{0.2}\right)}^{2}}$$
where *P* is productivity ranging from 1 to 2.5, and 1.5 is the range, *S* is susceptibility ranging from 0 to 0.2, and 0.2 is the range. Larger D value indicates higher risk.

## 3. Results

### 3.1. Sales Data Analysis

In total, more than 2.38 million individual landing records at Nangfangao and Hsinkang fishing ports from 2001–2021 were used in this study. The sales records of Nanfangao (northeastern Taiwan) and Hsinkang (southeastern Taiwan) fishing ports showed that shark landings ranged from 4738 tons in 2007 to 1404 tons in 2012 and maintained stability at around 1702 tons after then (Figure 2). Of this, the blue shark (27%), shortfin mako (20%), and bigeye thresher (16%) were the top three species in shark landings. The blue marlin (44%), swordfish (25%), and white marlin (14%) were the major billfish species landed at Nanfanao fishing port, while the white marlin (36%), sailfish (34%), and blue marlin (19%) dominated the billfish landings at Hsinkang fishing port. As for tuna landings, the yellowfin tuna was the top one for both fishing ports, followed by the Pacific bluefin tuna, the bigeye tuna in Nanfanao, and the bigeye and albacore in Hsinkang fishing port.

The sales records of Nanfangao fishing port indicated sharks had the highest catch in number followed by tunas and billfishes in 2001–2010; tunas had the highest catch in number followed by sharks and billfishes since 2011. The catch in number at Hsinkang fishing port was dominated by billfishes, followed by tunas and sharks. Shark catch in number of the two fishing ports ranged from 128,397 in 2006 to 27,008 in 2012. Of which, the blue shark (41%), shortfin mako (16%), and scalloped hammerhead (11%) were the top three species in shark landings. The catch in number of tunas ranged from 19,454 in 2005 to 47,091 in 2016. Of which, the yellowfin tuna comprised 76%, followed by bigeye tuna (13%) and albacore (8%). The billfish catch in number ranged from 19,836 in 2021 to 69,240 in 2010. Of which, the sailfish (45%), blue marlin (20%), and swordfish (16%) were the major billfish species landed.

### 3.2. Life History Parameters and Biological Characteristics

#### 3.2.1. Length

Asymptotic length (L_∞_): The largest L_∞_ of sharks was 422 cm TL for the bigeye thresher, and the smallest L_∞_ was 210 cm TL for sandbar shark; the maximum L_∞_ of tunas was 366.7 cm FL for the Pacific bluefin tuna, and the smallest L_∞_ was 103.5 cm FL for albacore tuna; the maximum L_∞_ of billfishes was 421.8 cm EFL for the blue marlin, and the smallest L_∞_ was 277.4 cm EFL for striped marlin (Appendix A).

Maximum observed length (L_max_): The largest L_max_ of sharks was 422 cm TL for the bigeye thresher, the smallest L_max_ was 210 cm TL for sandbar shark; the largest L_max_ of tunas was 252 cm TL for the Pacific bluefin tuna, and the smallest L_max_ was 101 cm FL for albacore tuna; the largest L_max_ of billfishes was 324.5 cm EFL for the blue marlin, and the smallest L_max_ was 191 cm EFL for striped marlin (Appendix A).

Size at birth (L_b_): The largest L_b_ of sharks was 174 cm TL for the bigeye thresher, and the smallest L_b_ was 45 cm TL for blue shark. No L_b_ information for tunas and billfishes was available as they were oviparous species (Appendix A).

Mean size at maturity (L_m_): The largest L_m_ of sharks was 336.6 cm TL for the bigeye thresher and the smallest L_m_ was 172.5 cm TL for sandbar shark. The largest L_m_ of tunas was 190 cm FL for the Pacific bluefin tuna and the smallest L_m_ was 83 cm FL for albacore. The largest L_m_ of billfish was 194.5 cm EFL for the white marlin and the smallest L_m_ was 143.54 cm EFL for sailfish (Appendix A).

#### 3.2.2. Age

Maximum age (A_max_): The largest A_max_ of sharks was 50 years for the dusky shark, the smallest A_max_ was 11.6 years for the scalloped hammerhead shark; the largest A_max_ of tunas was 26 years for the Pacific bluefin tuna, and the smallest A_max_ was 7.7 years for yellowfin tuna; the largest A_max_ of billfishes was 14 years for the blue marlin, and the smallest A_max_ was 6 years for striped marlin (Appendix A).Age at maturity (A_m_): The largest A_m_ of sharks was 16.4 years for the dusky shark, the smallest A_m_ was 4.7 years for the scalloped hammerhead shark; the largest A_m_ of tunas was 8 years for the Pacific bluefin tuna, and the smallest A_m_ was 2.4 years for yellowfin tuna; the largest A_m_ of billfishes was 7.4 years for the blue marlin, and the smallest A_max_ was 4.8 years for striped marlin (Appendix A).

#### 3.2.3. Other Life History Parameters

Fecundity/litter size: The largest litter size for sharks was the scalloped hammerhead of 30, and smallest was two for bigeye and pelagic thresher. The largest batch fecundity of tunas was the Pacific bluefin tuna of 5.8–25.2 million eggs, and the smallest was albacore of 0.94 million eggs; the largest of billfishes was the blue marlin of 6.94 million and the smallest was sailfish of 1.30 million (Appendix A).Reproduction mode: Carcharhinidae species, blue shark, and hammerhead sharks are viviparity, bigeye and pelagic thresher, and shortfin mako sharks are aplacental viviparity. Tunas and billfishes are oviparity.Trophic position (Tp): The Tp derived from the Ecopath [34] were as follows: 4.16 for Carcharhindae, 3.99 for blue shark, and blue marlin, 3.94 for swordfish, 3.89 for other billfishes, 3.78 for yellowfin tuna, and bigeye tuna, 3.75 for other tunas. The Tp of all the species was greater than 3.0, which was in high trophic position.

### 3.3. ERA

The shortfin mako and dusky shark had the lowest productivity index value (*p* = 1.00) which corresponded to the lowest productivity, followed by the bigeye, pelagic thresher and smooth hammerhead (*p* = 1.14), and the striped marlin and yellowfin tuna had the highest productivity index value of 2.43 (highest productivity) (Table 3). The yellowfin tuna had the highest susceptibility value (S = 0.1864), followed by the blue shark (S = 0.1344), sailfish (S = 0.1053), shortfin mako shark (S = 0.0613), and blue marlin (S = 0.0563). The oceanic whitetip shark had the lowest susceptibility value of 0.0009 due to its smallest catchability (0.0013) (Table 4).

The shortfin mako, dusky shark, and yellowfin tuna were the top three species with the highest ecological risk with D = 1.0459, 1.0028, and 0.9330, respectively. While the striped marlin, albacore, and bigeye tuna had the lowest ecological risk with D = 0.0564, 0.1612, and 0.2228, respectively (Table 5). Three groups can be categorized for the 20 species in this study as (1) the high ecological risk group of SMA and DUS, (2) the median ecological risk group including YFT, other shark species, and sailfish, and (3) the low ecological risk group of other tuna species and billfishes (Figure 3).

## 4. Discussion

This study assessed the impact on 20 pelagic fish species by the Taiwanese small-scale longline fishery in the western North Pacific using ERA. The results can provide useful information for implementing ecosystem-based management and prioritizing the management and conservation measures of pelagic species including tuna, billfishes, and sharks in this region.

### 4.1. Catch Data

In addition to Taiwanese longline fishing vessels, some of Japanese longline fishing vessels also operated in the western North Pacific Ocean. Future work, if possible, should refer to the study of Murua et al. [35], Cortés et al. [36], Murua et al. [28], and Cortés et al. [30] to include Japanese catch and observer data to improve the ERA of pelagic fishes in the western North Pacific. For example, only adult BFT was caught by the Taiwanese longline fishing vessels and examined in the present study, but both young and adult BFT were caught by Japanese fishing vessels. Better estimate of susceptibility of BFT can be obtained by examining the pooled data in the future.

### 4.2. Life History Parameters

Age estimation is the fundamental information of fish biology study and is one of the important life history parameters in stock assessment and population dynamics. The age of pelagic sharks used in this study was estimated based on the band pair counts of vertebral centra from the literature. As for tunas and billfishes used in this study, length-frequency analysis was used in growth parameter estimations for the yellowfin [37] and bigeye tuna [12]; otolith was used for the albacore [14], while the spine was used in the age estimation for the swordfish [38], blue marlin [39], sailfish [15], white marlin [40], and striped marlin [41]. The scale was used in the age estimate for the Pacific bluefin tuna [42]. The growth parameters derived from different hard parts of fish may be different which may lead to the bias of age-structure [43]. As the productivity was estimated based on the growth information of each species in this study, the bias and uncertainty of age estimation may affect the subsequent ERA results.

The A_mat_ and A_max_ and ERA of the scalloped hammerhead were based on a biannual deposition estimation [44] in this study. Although similar assumption was made by Anislado-Tolentino and Robinson-Mendoza [45] in Mexican waters, annual deposition was assumed by other authors in various regions of the Pacific Ocean [46,47,48]. Piercy et al. [49] reported A_max_ of 30.5 years, which was much larger than 11.6 years used in the present study. If annual deposition was used in the ERA as a different scenario, the risk ranking of the scalloped hammerhead was promoted to be 4 from 9 (Table 5). If the body weight variation trend and the IUCN Red List category were taken into account, this species was at the highest risk among 11 pelagic shark species [1].

The uncertainty existed in life history parameters used in this study, which may also be related to the small sample size. For example, the sample size was 188 for the age, growth, and reproductive biology study of the oceanic whitetip [50]. However, only two pregnant females were collected in that study. Thus, the small sample size may lead to the bias of little size and size at maturity estimations. Although the life history parameters used in this study were from the best available information, some of the references were published many years ago and may not represent current biological condition of fish. Fortunately, as the productivity was estimated by the mean rank of seven life history parameters, the productivity indicator did not change even if the uncertainty of life history parameters were taken into account. Future work should focus on collecting more specimens and using consistent ageing characteristics to get more accurate and updated estimates of life history parameters and improve the ERA results.

### 4.3. ERA

Two methods were commonly used to estimate the productivity of fish. The first method was based on the rank of life history parameters of each species, and the mean of rank was used as the productivity index (*p*) [32,51,52]. The *p* value can be estimated by another method using an empirical equation based on the reproductive strategy, size at maturity, and the maximum length [24,26,35]. The latter had higher uncertainty because *p* was estimated based on few life history parameters. Thus, the first method was used in estimation of *p* of each species in this study. The shortfin mako and dusky shark had the highest value of, *p* which was corresponding to the dusky shark in Australian waters [33].

The higher value of *p* indicated higher reproductive potential, which may experience lower risk. Overall, pelagic sharks are slow growing and late maturing, with extended longevity and low reproductive potential species. The shortfin mako and dusky shark mature at older ages than other pelagic shark species (Appendix A), and thus experienced higher risk than other shark species.

The intrinsic rate of population growth (r) has been used as productivity index for pelagic sharks [1,29,30,36]. Although the intrinsic rate of population growth was a better productivity index for pelagic sharks, this index might not be suitable for tuna and billfishes because demographic parameters of these oviparous species were difficult to estimate. Due to the remarkable difference in life history such as the fecundity (litter size) and longevity between sharks and other species, a rank-based index was believed to be a better choice as it can be applied to all species.

Susceptibility was commonly estimated by the multiplication of the availability, encounterability, selectivity, and post capture mortality in previous studies [29,30,32,36,51,52]. Due to the difficulty of obtaining the information of availability and encounterability, these two indices were replaced by the species-specific catchability estimated based on its mean percentage of catch in catch from 2001 to 2021 at Nanfangao and Hsinkang fishing ports. The long-term and large-size historical landing data (*n* > 2.38 million) can better describe the species-specific vulnerability to longline fishery. The low catchability may represent low abundance of fish species which is likely resulted from overfishing. Unfortunately, the susceptibility used in this study may not reflect the vulnerability of fishing for certain species after they were ban retention such as OCS and FAL as no discard data of these two species were available for susceptibility estimation. In addition, different sizes in fishing vessels and target species may affect the catchability. Future study should focus on collecting the information of number of hooks, main line length, branch line length, and species-specific capture depth from observer’s records and tagging results to estimate availability and encounterability to validate the catchability used in this study.

The selectivity estimated in this study was based on the ratio of age range of catch and the longevity. The age range was estimated by converting individual shark landing data (body weight) to length and then to age, which were used to estimate the minimum and maximum ages of each species. Uncertainty may occur in the two-step converting process. Those weights greater than the maximum weight in the literature were excluded in our analyses, yet these individuals may be pregnant females but could not be confirmed due to lack of sex information of sales records. Selectivity may be affected by the hunger condition of fish, bait type, hook type and size, and material of branch line (steel wire or monofilament) [53]. Since the selectivity was estimated by pooling the long-term data together, those effects aforementioned could be ignored. We believed that our estimates based on the best available data with large sample size were representative. As individual body weight data were not available for most blue sharks, the selectivity of blue shark being set as 1 was assumed due to its wide range of size at catch. We believe this assumption was reasonable.

ERA has been mainly applied to the bycatch species in the past. In this study, due to the remarkable difference in life history traits among sharks and other teleost species, the ranking method was used in assessing the vulnerability of 20 pelagic fish species. The ERA indicated that sharks have higher risk than tunas and billfishes, except the yellowfin tuna, which was comparable with the findings in the waters off eastern Taiwan [31], but some differences were found between the two studies. The SMA was identified as the highest ecological risk shark species in the present study but Lin et al. [31] suggested that FAL had the higher risk than BSH and SMA. As for other species, the present study concluded that YFT had the highest risk, followed by BLM and BUM, but Lin et al. [31] reported that BFT had the highest risk, followed by BET and BLM. The discrepancy may be due to different definitions and scores of productivity and susceptibility in these two studies. Previous ERA was based on reproductive strategy, the maximum size, size at maturity as productivity, and size at catch, and post capture mortality of animals as susceptibility from observer’s records [24,35,45]. However, high uncertainty existed in these studies due to the limited data available. In the present study, we used catchability, selectivity, and post capture mortality to estimate the susceptibility. If the availability and encounterability can be derived from observer’s data in the future, more robust results of ERA can be obtained.

### 4.4. Comparison with Single-Species Approach

Liu and Chen [3] used an age-structured demographic analysis in scalloped hammerhead stock assessment and concluded that the species could not withstand the long-term exploitation on juveniles. Liu et al. [4] and Tsai et al. [5] reported the stock status of pelagic thresher shark was overexploited using yield per recruit model (YPR) and stage-based demographic analysis. Several studies with various approaches indicated that the shortfin mako shark, bigeye thresher shark, and smooth hammerhead in the western North Pacific were overexploited [6,7,8,9,10,11,12]. As for tunas and billfishes, Liu [13] suggested that the fishing mortality (F) of the bigeye tuna in Taiwan waters was greater than the biological reference point (F_0.1_ and F_SSB30_) and was in full exploitation. Chung [14] evaluated the North Pacific bluefin tuna using a non-equilibrium production model and suggested a total allowable catch of 18,000 tons. Chen [15], Chiang [16] and Wang [17] conducted the stock assessment of the albacore, sailfish and swordfish in the North Pacific and concluded that the F of these species were at optimum levels.

Although the two approaches were based on different theories, the existing results of single species stock assessment of several species mentioned above were comparable with those derived from this study using the ERA method, suggesting that sharks have higher ecological risk than tuna and billfishes.

## 5. Conclusions

In the present study, the semi-quantitative ERA has prioritized the risk of 20 pelagic fish species in the western North Pacific. Sharks have higher ecological risk than tunas (except yellowfin tuna) and billfishes. The shortfin mako and dusky shark have the highest ecological risk; yellowfin tuna, other shark species, and sailfish have medium risk, while the striped marlin, and albacore tuna had the lowest risk. Although ERA cannot replace the conventional stock assessment methods that can produce solid management information on catch and effort, it can provide useful information for precautionary management measures. Rigorous management measures such as catch quota and size limit are recommended for the species at high or medium ecological risk and a consistent monitoring management measure is suggested for other species at low ecological risk. To improve the results, the ERA should be updated regularly based on best available information of the productivity and susceptibility of these species.

## Figures and Tables

**Figure 1 animals-12-02124-f001:**
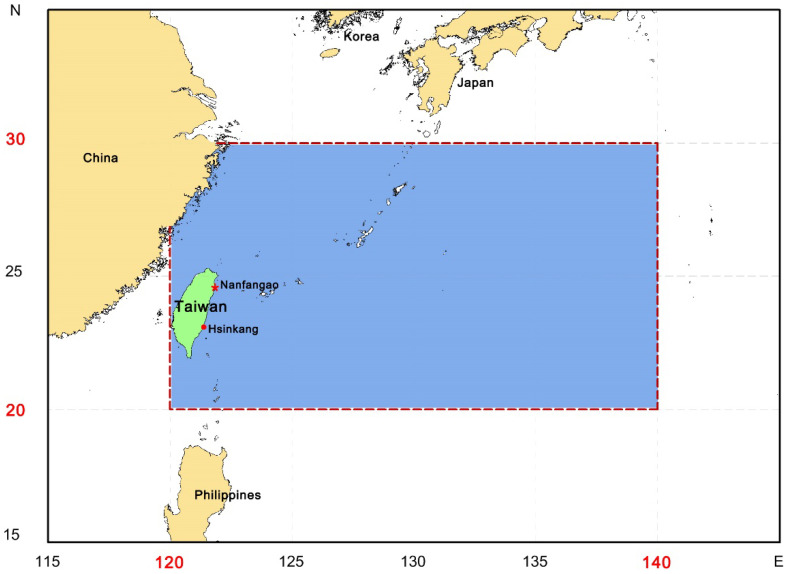
The study area (from 20° N to 30° N, 120° E to 140° E) in the present study. Red star is Nanfangao fishing port, red circle is Hsinkang fishing port.

**Figure 2 animals-12-02124-f002:**
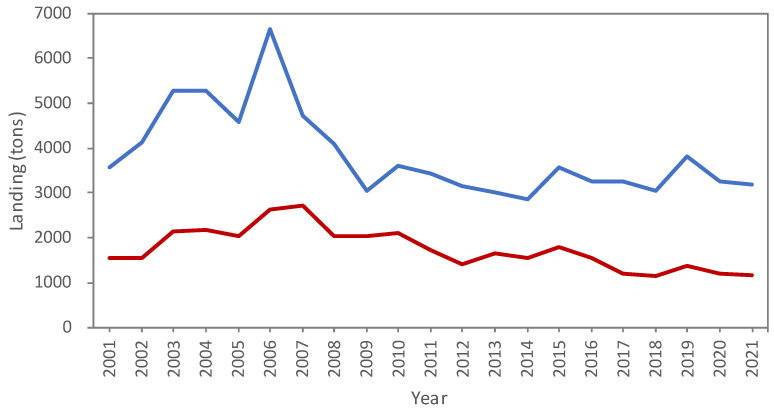
Annual landing at Nanfangao (blue line) and Hsinkang (red line) fishing ports from 2001 to 2021.

**Figure 3 animals-12-02124-f003:**
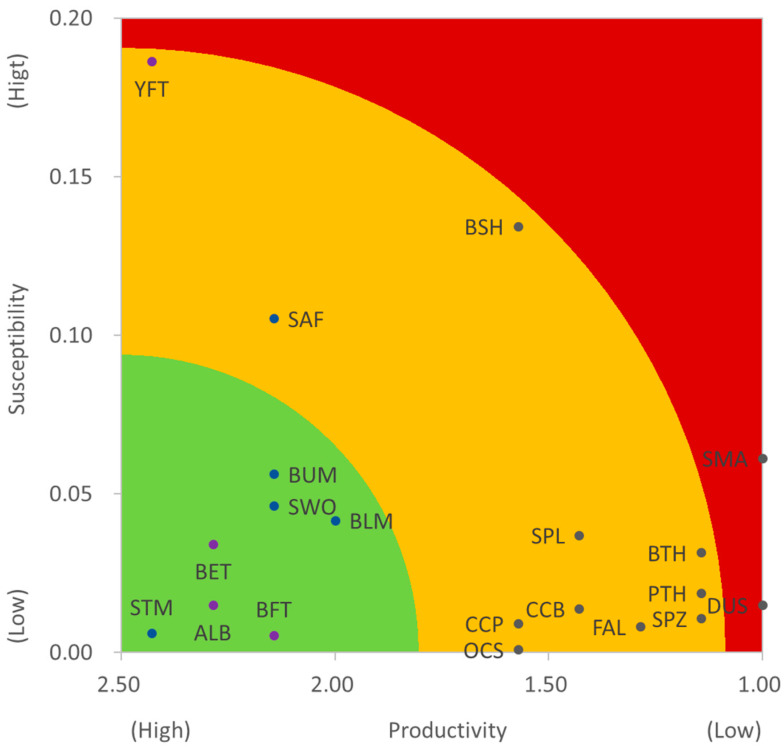
Productivity and susceptibility plot for the 20 pelagic fish species in the western North Pacific Ocean.

**Table 1 animals-12-02124-t001:** The 20 pelagic fish species in the western North Pacific Ocean examined in this study.

Scientific Name	Common Name	Species Code	IUCN Status
*Alopias pelagicus*	Pelagic thresher shark	PTH	EN
*A. superciliosus*	Bigeye thresher shark	BTH	VU
*Carcharhinus brevipinna*	Spinner shark	CCB	NT
*C. falciformis*	Silky shark	FAL	VU
*C. longimanus*	Oceanic whitetip shark	OCS	CR
*C. obscurus*	Dusky shark	DUS	EN
*C. plumbeus*	Sandbar shark	CCP	VU
*Isurus oxyrinchus*	Shortfin mako	SMA	EN
*Sphyrna lewini*	Scalloped hammerhead	SPL	CR
*S. zygaena*	Smooth hammerhead	SPZ	VU
*Prionace glauca*	Blue shark	BSH	NT
*Makaira nigricans*	Blue marlin	BUM	VU
*Istiompax indica*	Black marlin	BLM	DD
*Istiophorus platypterus*	Sailfish	SAF	VU
*Xiphias gladius*	Swordfish	SWO	NT
*Kajikia audax*	Striped marlin	STM	LC
*Thunnus alalunga*	Albacore tuna	ALB	LC
*T. albacares*	Yellowfin tuna	YFT	LC
*T. obesus*	Bigeye tuna	BET	VU
*T. orientalis*	Pacific bluefin tuna	BFT	NT

DD: data deficient, LC: least concern, NT: near threatened, VU: vulnerable, EN: endangered, CR: critically endangered.

**Table 2 animals-12-02124-t002:** Productivity cutoff scores for species attributes for the ERA method, and the attribute values into low, medium and high productivity categories.

Attribute	Low Productivity High Risk (R = 1)	Medium Productivity Median Risk (R = 2)	High Productivity Low Risk (R = 3)
Age at maturity (A_m_)	>15 years	5–15 years	<5 years
Length at maturity (L_m_)	>200 cm	40–200 cm	<40 cm
Maximum age (A_max_)	>25 years	10–25 years	<10 years
Maximum length (L_max_)	>300 cm	100–300 cm	<100 cm
Fecundity (F)	<100	100–20,000	>20,000
Reproductive strategy (R)	viviparity or aplacental viviparity	with parental care after spawning	without parental care after spawning
Trophic position (TP)	>3.25	2.75–3.25	<2.75

**Table 3 animals-12-02124-t003:** Productivity parameters by rank in the ecological risk assessment of the 20 pelagic species in the western North Pacific. The productivity indicator (P) reflects the risk of productivity.

Species	Rank							
Code	A_m_	L_m_	A_max_	L_max_	F	R	TP	P
PTH	2	1	1	1	1	1	1	1.14
BTH	2	1	1	1	1	1	1	1.14
CCB	2	1	2	2	1	1	1	1.43
FAL	2	1	1	2	1	1	1	1.29
OCS	2	2	2	2	1	1	1	1.57
DUS	1	1	1	1	1	1	1	1.00
CCP	2	2	2	2	1	1	1	1.57
SMA	1	1	1	1	1	1	1	1.00
SPL	3	1	2	1	1	1	1	1.43
SPZ	2	1	1	1	1	1	1	1.14
BSH	3	2	2	1	1	1	1	1.57
BUM	2	2	2	2	3	3	1	2.14
BLM	2	2	2	1	3	3	1	2.00
SAF	2	2	2	2	3	3	1	2.14
SWO	2	2	2	2	3	3	1	2.14
STM	3	2	3	2	3	3	1	2.43
ALB	3	2	2	2	3	3	1	2.29
YFT	3	2	3	2	3	3	1	2.43
BET	3	2	2	2	3	3	1	2.29
BFT	2	2	2	2	3	3	1	2.14

A_m_: age at maturity, L_m_: length at maturity, A_max_: maximum age, L_max_: maximum length, F: fecundity/litter size, R: Reproduction strategy, TP: tropic position.

**Table 4 animals-12-02124-t004:** Susceptibility parameters in the ecological risk assessment of the 20 pelagic species in the western North Pacific. The susceptibility (S) reflects the risk of fisheries development. The S is multiplied by three parameters, catchability, selectivity and post-capture mortality.

Species Code	Catchability	Selectivity	Post-Capture Mortality	S
PTH	0.0241	1.0000	0.7800	0.0188
BTH	0.0404	1.0000	0.7800	0.0315
CCB	0.0180	0.9970	0.7700	0.0138
FAL	0.0095	0.9960	0.8600	0.0082
OCS	0.0013	0.9350	0.7700	0.0009
DUS	0.0163	1.0000	0.9200	0.0150
CCP	0.0113	0.9480	0.8600	0.0092
SMA	0.0666	1.0000	0.9200	0.0613
SPL	0.0461	0.9680	0.8300	0.0370
SPZ	0.0133	0.9490	0.8500	0.0107
BSH	0.1702	1.0000	0.7900	0.1344
BUM	0.0611	0.9210	1.0000	0.0563
BLM	0.0429	0.9730	1.0000	0.0417
SAF	0.1368	0.7700	1.0000	0.1053
SWF	0.0502	0.9210	1.0000	0.0462
STM	0.0158	0.3830	1.0000	0.0061
ALB	0.0231	0.6460	1.0000	0.0149
YFT	0.2089	0.8920	1.0000	0.1864
BET	0.0352	0.9700	1.0000	0.0342
BFT	0.0089	0.6080	1.0000	0.0054

**Table 5 animals-12-02124-t005:** Ecological risk assessment results of the 20 pelagic species in the Northwest Pacific Ocean.

Species Code	P	S	ERA	Risk Ranking
PTH	1.1429	0.0188	0.9096	6
BTH	1.1429	0.0315	0.9184	5
CCB	1.4286	0.0138	0.7176	10
FAL	1.2857	0.0082	0.8106	8
OCS	1.5714	0.0009	0.6191	12
DUS	1.0000	0.0150	1.0028	2
CCP	1.5714	0.0092	0.6207	11
SMA	1.0000	0.0613	1.0459	1
SPL	1.4286	0.0370	0.7379	9
SPZ	1.1429	0.0107	0.9064	7
BSH	1.5714	0.1344	0.9138	4
BUM	2.1429	0.0563	0.3686	15
BLM	2.0000	0.0417	0.3932	14
SAF	2.1429	0.1053	0.5779	13
SWO	2.1429	0.0462	0.3317	16
STM	2.4286	0.0061	0.0564	20
ALB	2.2857	0.0149	0.1612	19
YFT	2.4286	0.1864	0.9330	3
BET	2.2857	0.0342	0.2228	18
BFT	2.1429	0.0054	0.2396	17

## Data Availability

The data used in this study can be found in Appendix A.

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
