# Peer review of "Assessment of the Impact on 20 Pelagic Fish Species by the Taiwanese Small-Scale Longline Fishery in the Western North Pacific Using Ecological Risk Assessment"

_animals, 2022, doi:10.3390/ani12162124_

Round 1
Reviewer 1 Report
The paper by Liu et al applies Ecological risk assessment (ERA) to 20 species of pelagic fish species by the small-scale longline fisheries in the western North Pacific Ocean. This is an interesting contribution as many of these species, especially sharks, are sensitive to decline around the world and their monitoring is crucial. I appreciated the paper, but in my opinion some of the methodologies followed and / or applied are not clear.
I believe that after a better detail on these aspects and on the comments below, the paper is worth publishing.
- line 39: Are pelagic sharks a target species of fishing boats, or are they a bycatch?
- line 63: change “billfises” to “billfishes”
- line 98: I suggest to use the word “aim” instead of “objective”
- Table 1: it is better to use, as in the text, “Pelagic thresher” and “Bigeye thresher” instead of ““pelagic thresher shark” and “Bigeye thresher shark”
- line 289: authors say that “The age of pelagic sharks was estimated based the band pair counts of vertebral centra in this study”. It is not clear to me, therefore, whether the authors collected the vertebrae and carried out the counts of the growth rings. If so, they must necessarily enter the protocol used in the "materials and methods" section. The same goes for the spines and otoliths in the following lines. If it is only data taken from the literature, this must be clearly explained, because otherwise it creates confusion in the reader
- Supplemental Table 1. In my opinion, I consider the information contained in the table to be very interesting. The authors may consider including it in the paper
Author Response
- line 39: Are pelagic sharks a target species of fishing boats, or are they a bycatch?
Response: Some of the fishing vessels seasonal target sharks (Oct. – March) but others are bycatch. We have added citation to this sentence.
- line 63: change “billfises” to “billfishes”
Response: The typo has been corrected.
- line 98: I suggest to use the word “aim” instead of “objective”
Response: As “aim” and “objective” have similar meanings, we prefer to keep the same.
- Table 1: it is better to use, as in the text, “Pelagic thresher” and “Bigeye thresher” instead of ““pelagic thresher shark” and “Bigeye thresher shark”
Response: To distinguish with other species (tuna and billfishes), we believe “Pelagic thresher shark” and “Bigeye thresher shark” are more appropriate terms.
- line 289: authors say that “The age of pelagic sharks was estimated based the band pair counts of vertebral centra in this study”. It is not clear to me, therefore, whether the authors collected the vertebrae and carried out the counts of the growth rings. If so, they must necessarily enter the protocol used in the "materials and methods" section. The same goes for the spines and otoliths in the following lines. If it is only data taken from the literature, this must be clearly explained, because otherwise it creates confusion in the reader
Response: This sentence has been changed as “ The age of pelagic sharks used in this study was estimated based on the band pair counts of vertebral centra from the literature”.
- Supplemental Table 1. In my opinion, I consider the information contained in the table to be very interesting. The authors may consider including it in the paper
Response: Supplemental Tables 1-3 are the life history parameters of 20 species from the literature not derived from this study. So, we think that these tables are more appropriate treated as supplemental information.
Reviewer 2 Report
The manuscript entitled: "Assessment of the impact on 20 pelagic fish species by the Taiwanese small-scale longline fishery in the western North Pacific using ecological risk assessment" provides an ecological risk analysis result on many pelagic fish species in the Western North Pacific area. I have to say the manuscript is not acceptable in its current form. Since the quality of the study is poor for publication in a peer review journal. In the introduction part, the sentences were written without logical thinking. In the methods part, as the author admitted, massive out-of-date data and parameters were used in this study, making the result with high uncertainty. Also, detailed information was missed in the manuscript. Moreover, the data analysis techniques they used in this study did not strictly follow the references (especially for the “catchability” and “selectivity” value estimation process). In the discussion part, most of the discussion is like an extension of the methods they used. The authors need to rewrite the whole manuscript before resubmitting it. This manuscript might be more suitable for a conference report, but not a peer review paper. I attached my comments in the PDF file.

Author Response
Response to Reviewer 2
Title:delete "Taiwanese". Since the Japanese fishery boat also working in this area (there maybe also have other party's fishing boat). the state of those fishes were not only impact by Taiwanese fishery.
Response: The Taiwanese longline fishing vessels are the major players in the study area given some minor efforts may be from Japan or other countries. As fisheries information of other countries in this area is not available, I believe the data used in this study can represent the major fishing impact on these pelagic species.
L 32, The annual landing data were from sales records of the two fishing ports. So, no citation is needed.
L 36, Citation missed everywhere, please double-check all the words.
Response: We have checked the text and added citations in several places.
L 53-76, Is there any necessary reason to list the stock assessment method here?
There was no explanation and background introduction on this aspect.
Response: Single species stock assessment is a conventional way to estimate the vulnerability of a species caused by fisheries. However, this approach can not identify the ranking of ecological risk for a group species which caught by the same fishery such as longline. We have pointed out this limitation and shorten the wordings and moved most text to Discussion section to make comparison with ERA results as suggested. The motivation of using alternate approach such as ERA to achieve the goal is also mentioned in this revised version.
L 78, The author did not give a clear background to show why we need ecological risk assessment.
Response: As we mentioned above, we expressed the limitation of single species stock assessment and the need of ecosystem approach for better understand the vulnerability of 20 pelagic fish species. May be the text is not clear, we have revised the text to give a clearer background of the need of ERA.
L 96-98, Only Taiwanese fishery in this area? At least, author should provide the proportion of Taiwanese fishery in the study area.
Response: Although we believe the Taiwanese longline fishery is the majority in the study area, as the fishery data of other countries are not available (either from the literature, published fisheries statistics, or ISC (International Scientific Commission) data base), we can not estimate the exact proportion of Taiwanese fishery in the study area.
L 112, A lot of detailed information were missed here. For example:
- where is the data from? logbook? yearbook? survey program?
- why choose these two ports?
- what are the fishery gear composition and the technical parameter of those gears?
Response: The paragraph 2. 1. Study area and species has most information needed. 1. These data were from sales records which was mentioned in the next sentence (L. 113-115).
- Most fish caught (~80%) by the Taiwanese longline fishery in the study area were landed in the two fishing ports. So, the landing data of these two ports were chosen for analysis.
- Longline is the major fishing gear in the study area thus its impact on pelagic fish was evaluated in this study.
L 119, “citation”
Response: The citation “Supplemental Table 1” has been added.
L 121-123, Literature data? survey data? state explicitly.
Response: This mean weight was from unpublished on board scientific observer records. So, no citation can be added. We have revised this sentence.
L 130, Delete. we can get this information in line 129
Response: Instead of deleting this line, we deleted “Table 2” in Line 129.
Table 2, criterion of such scoring should be given
Response: The criterion of scoring was developed in this study based on the life history information of 20 species. The criteria of scoring from other study are not necessarily be applied to this study as the species and study area are different.
The authors tried to used catch composition data represents the availability and encounterability. I don't think those items are same meaning. Availability and encounterability are relate with the fleet number, active area and other issues. While the catch composition can not include such information.
Compare to the references, the author used totally different parameters to estimate the ecological risk, therefore, the ecological theory need to be added.
Response: Because the availability and encounterability are difficult to estimate we used catch composition to represent the fishing impact on fish species. The catch composition is a combined result of horizontal and vertical overlap of fish distribution and fishing gear (availability and encounterbility), and the probability being caught. As the catch composition was from more than 2.38 million individual fish from 2001-2021, even the meaning may be not the same as the literature, we believe the large data set can best describe the fishing impact on 20 pelagic fish species in the western North Pacific.
L 143, There is no detailed explanations on how estimated the selectivity.
Response: Selectivity used in this study was based on Webb et al. (2017). We have added the formula and citation in this version.
Selectivity =(Amax_c – Amin_c)/Amax, where Amax_c is the maximum age at catch, Amin_c is the minimum age at catch, Amax is the maximum age (longevity).
Fishing gear in the Cortes's study include longline, purse seiner and gillnet.
Can we assume that the post-mortality were same in this study and the reference's study?
Response: Given the post-mortality information is very limited in the literature, we assume the post-mortality in this study is similar to Cortes study even the fisheries are different.
L 150, The authors did not give detailed information on this equation.
What 2.5, 0, 1.5, 0.2 mean in this equation?
Those parameters were also different when compare with the reference.
The authors need explain it in detail.
Besides, reference [26] is slides, and is not the original references, it is better to cite "Cortés, E., Arocha, F., Beerkircher, L., Carvalho, F., Domingo, A., Heupel, M., Holtzhausen, H., Santos, M.N., Ribera, M., Simpfendorfer, C., 2010. Ecological risk assessment of pelagic sharks caught in Atlantic pelagic longline fisheries. Aquat. Liv. Resour., 23: 25-34. "
Response: As the range of productivity and susceptivity are in different scale (productivity ranging from 1 – 2.5 with the range of 1.5, and susceptivity ranging from 0 – 0.2 with the range of 0.2), we can not use the original formula to calculate Euclidean distance (D) because D is mainly affected by productivity due to it is 10 folds of susecptivity. A modified method with the standardization of the two axes was used to estimate the distance from origin. We have added some text to describe the meaning of the formula. We have changed the reference of Cortes et al. (2010) [30] based on the suggestion.
L 152, Are there any more detailed information for the range of D. ie, How large is large?
Response: The D value is the distance from the origin (the highest productivity and lowest susceptivity). Therefore, as the two axes have been standardized (as mentioned above), the larger the value of D, the higher the risk is.
L 156-158, 3.1 table or figure should be given
Response: We have added one figure (Figure 2) to show the annual landings of the two fishing ports.
L 180, I noticed that most length parameter were not measured in the same year. some of the references publish 20 yeas ago. The potential impact of history parameter on the estimation of this study should be addressed.
I strongly suggest to add a uncertainty analysis and evaluate the impact of such parameter uncertainty on the result.
Response: The life history parameters used in this study were based on the best information available. As no new information was available, some of the studies were based earlier results. The productivity was estimated by the mean rank of seven life history parameters. We have tried to included the uncertainty of the life history parameters in P estimation for those species with confidence intervals in their life history parameter estimations. However, the rank (R=1, 2, 3) of each life history parameter is the same even the uncertainty is taken into account.
L 255, “were”
Response: “was” is used in this version.
4.1 and 4.2
Most of the content can move to the methods part.
This is just a detailed explanation on how to find, set and choose each parameters.
During those description, we can find a lot of shortage in the data collection and parameter estimation.....
Response: We have moved most of the text to M & M section.
L 327, “cite”
Response: This information can be found in Supplemental Table 2. We had added (Supplemental Table 2) in the text.
L 352-354, The selectivity measure way is also wired. Anyway, it is not the way used in fish stock assessment. and not the way used in reference. See reference [27]:
"Selectivity is the proportion of the individuals captured by the fishing gear provided that they are encountered."
Response: As our response of L 143, the selectivity used in this study was from Webb et al. (2017) which was different from that used in stock assessment. The definition of selectivity in this study is the proportion of age range of individuals captured by the fishing gear in relation to the longevity.
4.4
I didn't see any linkage between 4.4 and other part of this manuscript.
What is the necessity to write such part?
It might be better if the authors compare the ERA result with the stock assessment result, and discuss the different of these two kind of results.
Response: 4.4 has been removed. We have moved partial text of the single species stock assessment in Introduction section to 4.4 and make comparison with ERA results.
Reviewer 3 Report
Dear Authors,
I have read and checked your MS and added some suggestions within the file. I think that this study deserves to be published.
Kind regards

Author Response
Response to Reviewer 3
L 33-34, Why don't you mention the values of 2006 before 2014 in line 33 and the values of 2007 before those of 2018 in line 34?
Response: We have revised the text based on chronical series as suggested.
L 62, please, write it in italic
Response: This text has been removed in this version.
L 100-102, It sounds as a conclusion. I would move this sentence in the conclusion section
Response: “It is hoped that” has been added to the beginning of this sentence.
L 125, Please, add "method" after "ERA"
Response: “method” has been added.
Table 1, Please, add the IUCN status for each species.
Response: We have added one column in Table 1 to show the IUCN status for 20 species. “Thunnus” has been replaced by “T.” for YFT, BET, and BFT.
L 158, See lines 33-34
Response: The description is based on chronicle. “the” has been removed.
L 171, See lines 33-34
Response: The description is based on chronicle as L 33-34 and L 158.
L 180-183, Please, move it in M&M section
Response: This paragraph has been moved to the end of the first paragraph of M & M.
L 226, “ERA” is use in this version.
Table 3, Please, change as "Productivity parameters"
Response: “Productivity parameters” is used in this version.
L 290-293, If possible, it would be useful to have an example image of one vertebrae, one otolith and one spine with band counts as a Supplementary Material.
Response: The age study of 20 species based on vertebral, otolith, and spine were from the literature. We don’t have these images and the images of these hard parts can be found from these articles.
L 398,
Response: “In the present study” has been moved to the beginning of the sentence.
Round 2
Reviewer 1 Report
Authors have addressed all comments and made changes that improved the paper. In my opinion now it is suitable to be published
Reviewer 2 Report
The authors responded appropriately to all of the reviewers' questions. I have no more suggestions